# Trophic niche variation across the pan-Arctic coastal continuum

Nathan D. McTigue [1,2]*, Katrin Iken[1], Ashley Ehrman[3], Bodil A. Bluhm[4], Guillaume Bridier[5], Rolf Gradinger[4], Joanna Legeżyńska[6], Maeve McGovern[7], Bailey McMeans[8], Frédéric Olivier[9¤], Amanda Poste[7], Paul E. Renaud[10], Virginie Roy[11], Janne E. Søreide[12], Maria Włodarska-Kowalczuk[6], Kenneth H. Dunton[2]

1 College of Fisheries and Ocean Sciences, University of Alaska Fairbanks, Fairbanks, Alaska, United States of America, 2 Marine Science Institute, The University of Texas at Austin, Port Aransas, Texas, United States of America, 3 Fisheries and Oceans Canada, Winnipeg, Manitoba, Canada, 4 Department of Arctic and Marine Biology, UiT The Arctic University of Norway, Tromsø, Norway, 5 Institut des Sciences de la Mer de Rimouski, Université du Québec À Rimouski, Rimouski, Quebec, Canada, 6 Institute of Oceanology Polish Academy of Sciences, Sopot, Poland, 7 Norwegian Institute for Water Research, Tromsø, Norway, 8 Department of Biology, University of Toronto Mississauga, Ontario, Canada, 9 Biologie des Organismes et Ecosystèmes Aquatiques, UMR MNHN/SU/UA/CNRS/IRD, Paris, France, 10 Akvaplan-niva, Tromsø, Norway, 11 Maurice Lamontagne Institute, Fisheries and Oceans Canada, Mont-Joli, Quebec, Canada, 12 Arctic Marine Biology and Ecology, University Centre in Svalbard, Longyearbyen, Norway

¤Current address: University of Brest, CNRS, IRD, IUEM, Plouzané, France
* mctigue@utexas.edu

## Abstract

We analyzed stable carbon and nitrogen isotope values ($\delta^{13}$C and $\delta^{15}$N, respectively) for pan-Arctic coastal primary producers and consumers to detect large-scale regional trends both temporally and spatially. To facilitate comparison, we grouped coastal habitats into fjords, lagoons, shelves, and straits as four "coastscapes". We gathered over 12,000 rows of data collected over 24 years (between 1999 and 2022) from 34 different field campaigns across the coastal Arctic (63 to 81°N and 177°W to 33°E). Our goal was to examine the isotopic patterns in pelagic and sediment particulate organic matter (pPOM and sPOM, respectively) and four consumer groups (deposit feeders, opportunists/scavengers, predators, and suspension feeders) among the four coastscapes. We found that despite the enormous spatial range of data, both pPOM and sPOM became 2.1‰ and 2.2‰ more $^{13}$C-depleted per decade, respectively, with parallel decreases in the $\delta^{13}$C values in consumers. The significant decrease is likely attributed to the increased contributions of $^{13}$C-depleted terrestrial organic matter across the Arctic coasts from freshwater inputs and coastal erosion in concert with diminishing sea ice that supports sympagic microalgae. Across all Arctic coastscapes, consumer groups exhibited overlapping isotopic composition, notably with wide $\delta^{13}$C ranges that indicated assimilation of multiple organic matter sources, including terrestrial organic matter, organic matter derived from marine phytoplankton and sea ice algae, macroalgae, and potentially benthic microalgae or degraded

**Data availability statement:** All data used for this study are publicly available at the Environmental Data Initiative under the DOI: https://doi.org/10.6073/pasta/48570e835463e405c707a4905c9d172d.

**Funding:** This research was funded through the 2017-2018 Belmont Forum and BiodivERsA joint call for research proposals, under the BiodivScen ERA-Net COFUND program, and with the funding organizations National Science Foundation (#1906726), Research Council of Norway (#296836/31406), and National Science Center, Poland (NCN #31382). The work was also partially funded by the Beaufort Lagoon Ecosystems LTER (NSF #1656026 and #2322664) and the Arctic Marine Biodiversity Observing Network (NOAA #NA19NOS0120198, ONF #N00014-22-1-2793, NASA #80NSSC22K1780). BB contributed as part of the Kitikmeot Sea Science Study (K3S) with funding provided by Fisheries and Oceans Canada, Polar Knowledge Canada, and the Arctic Research Foundation. GB and FO were funded by MarineBASIS and the Greenland Monitoring Program. The funders had no role in study design, data collection and analysis, decision to publish, or preparation of the manuscript.

**Competing interests:** The authors have declared that no competing interests exist.

organic matter. This consistent pattern across coastscapes provides evidence of the trophic plasticity possessed by Arctic consumers, how coastal food webs respond to climate warming, and the signature of terrestrialization imprinted on the pan-Arctic coastal isoscape.

## Introduction

The Arctic Ocean contains only 4.3% of the world's ocean area but hosts a disproportionate 34% of the world's coastline [1,2]. This expanse of coastal ecosystems lining the Arctic Ocean is subjected to the impacts of warming-induced changes from both adjacent terrestrial and marine ecosystems. The Arctic is warming at a rate four times the global average [3], which is catalyzing a variety of ecosystem-level changes on land and in the ocean that ultimately impact the coastal environment. The Arctic terrestrial landscape is undergoing warming-driven increases in precipitation [4], river discharge [5], and subsurface runoff [6], widespread thawing of permafrost [7], and accelerating rates of coastal erosion [8]. Meanwhile, accelerated decreases in ice extent, duration, and thickness have manifested in the Arctic [9,10]. The resulting longer fetch of open water promotes more frequent storms [11] that generate waves that exacerbate coastal erosion, increase sediment load, and/or deliver upwelled nutrients with the potential to enhance primary production [8,12]. The combined changes occurring in coastal ecosystems directly affect human populations and are likely more pronounced than those occurring in the open waters of the Arctic Ocean [13].

The downstream effects of these alterations in terrestrial and oceanic processes directly impact coastal food webs through changes in autochthonous primary production and the delivery of allochthonous organic matter. For example, additional freshwater inputs can increase stratification that restricts nutrient exchange into the euphotic zone limiting primary production [14]. But increased freshwater discharge paired with permafrost thawing also delivers terrestrial organic matter that is assimilated by coastal consumers [15,16], inorganic nutrients that can fuel primary production where sufficient light persists [17], and advected freshwater primary producers [18]. The change in the timing of sea ice retreat will impact the abundance and species composition of sea ice algae, and will likely favor phytoplankton production [19].

As the de-icing of the Arctic continues, increases in net primary production are disproportionately higher than increases in open water area [20], implying new terrigenous sources of nutrients are fueling coastal photosynthesis [17]. Conversely, increases in atmospheric cloudiness [21] and coastal turbidity [22] in the Arctic attenuate light transmission into coastal waters, which limits primary production. The interplay between decreasing sea ice extent, more days of open water, inputs of additional nutrients, and various factors affecting photosynthetically available radiation (PAR) results in unknown outcomes for coastal primary producers in terms of community composition, timing, and net production [23]. Given the plethora of environmental changes that are impacting primary production and carbon-source diversity, examining food web structure across time and space can signify how these changes are integrated at an ecosystem level.

Stable isotopes provide a useful and proven lens to examine trophic ecology. Hinging on the tenet that stable isotope ratios of consumers are related to those of their food sources, it is possible to use these biomarkers to calculate food source contribution and trophic position at the species level [24]. Conceptual advances in stable isotope ecology have embraced ecological niche theory [25] with quantitative devices to describe the "isotopic niche" by plotting two-dimensional stable isotope space (e.g., a $\delta^{13}$C-$\delta^{15}$N biplot), describing data behavior using spatial metrics [26], and calculating spatial overlap between groups [27,28]. This concept has persisted while the analytical applications continue to evolve with Bayesian implementation [29,30] and new metrics that are independent of sample size [31]. Patterns in stable isotope distribution (isoscapes) reveal trends in baseline resource availability [32] that can be used for applications such as ecogeochemical tracking of marine animal migration patterns [33] or regional variation in freshwater inputs [34]. Spatial (pan-Arctic) and temporal (decadal) isoscapes have been documented across the Arctic for pelagic particulate organic matter [35]. Moreover, there is a contemporary emphasis on integrating biogeochemical processes at the pan-Arctic scale to understand the future Arctic Ocean [36], including expansion of macroalgae [37], changing paradigms of under-ice blooms [38], and Arctic Basin carbon budgets [39]. Examining consumer isoscapes at the pan-Arctic, multi-decadal scale is a logical next step to continue along this scientific trajectory.

Here, stable carbon ($\delta^{13}$C) and nitrogen isotope ($\delta^{15}$N) values for pan-Arctic coastal primary producers and consumers were employed to detect potential large-scale regional trends both temporally and spatially. Upon this framework, we also take into consideration that the Arctic coastline is typified by a variety of features, including fjords, rocky shorelines, bays, gulfs, sounds, shelves, eroding bluffs, lagoons, straits, archipelagoes, deltas, and rivers fed by meteoric water and glaciers. To facilitate comparison, a reductionist approach was applied to the continuum of coastal habitats, grouping them by commonality related to food web end-member availability, which we call "coastscape" [40]. Specifically, we ask: what are the temporal and spatial patterns in isotopic niches of consumers among Arctic coastal ecosystem habitat types (coastscapes) and geographic regions? We test the following null hypotheses: (1) stable isotope values of end-members and consumers do not change on a multi-decadal scale, (2) isotopic niches of consumers exhibiting the same feeding habit do not differ among pan-Arctic coastscapes, and (3) isotopic niches of consumers from a common coastscape do not differ regionally across the Arctic. By addressing these hypotheses, we can evaluate the relative importance of different organic matter sources to consumers and assess if the delivery of terrestrial organic matter sources to coastal food webs across the Arctic is changing across time and space. Synthesizing this information at the pan-Arctic level is a novel approach to investigate trophic niche variation and explore the ecological plasticity of consumers at the land-sea-ice interface of the coastal Arctic where multiple environmental changes are occurring in tandem.

## Methods

### Data collation

A non-exhaustive survey was conducted to assemble stable carbon ($\delta^{13}$C) and nitrogen ($\delta^{15}$N) data from coastal habitats from 63 to 81 °N and 177 °W to 33 °E. Data were gathered from 34 sampling campaigns between 1999 and 2022, including published data [15,32,41–67] and other unpublished sources (see S1 Table for full details). From this, 12,189 rows of data were aggregated that satisfied the requirement of including a collection date, latitude and longitude of collection site, collection depth, both $\delta^{13}$C and $\delta^{15}$N values, and a taxonomic identification. Julian date was determined for the dataset using the earliest date of collection as the origin (i.e., zero). Stable isotope ratios from all collated data are represented in the δ notation as parts per thousand (‰) determined by the equation $\delta X = ((R_{sample}/R_{standard}) - 1) \times 1000$, where X is either $^{13}$C or $^{15}$N, and R is the ratio $^{13}$C/$^{12}$C or $^{15}$N/$^{14}$N, respectively. International standards, Pee Dee Belemnite for $\delta^{13}$C (PDB: USGS 24), and atmospheric air for $\delta^{15}$N (IAEA-N-1 and 2,) were used to determine $R_{standard}$.

## Defining coastscapes

Following the conceptual framework devised by international efforts integrating science with Indigenous and local knowledge [40], we identified common physiographic features within the coastal Arctic ecosystem continuum that ultimately influence food source availability (Table 1). Collection sites were grouped based on their physical connection to other bodies of water, potential interaction with the coastline, allochthonous organic matter delivery through glacial or riverine inputs, and macroalgal subsidy presence. Accordingly, four "coastscapes" were identified a priori as either fjord-like, lagoon-like, shelf-like, or strait-like habitats (Table 1) to facilitate subsequent analysis at the pan-Arctic scale. For simplicity, coastscapes are referred to hereafter as fjord, lagoon, shelf, or strait.

## Regional Arctic sector designation

We are aware that the coastscape coverage in this analysis was confounded by regional distributions and spatial autocorrelation (see Fig 1). For example, fjords are absent from the Beaufort Sea and Northern Bering/Chukchi Seas regions, and straits are common in the Canadian Arctic Archipelago but only rarely represented in the rest of the study area. Therefore,

**Table 1. Heuristics developed to identify coastscapes in the coastal Arctic habitat continuum.**

| Coastscape characteristic | Fjord | Lagoon | Shelf | Strait |
|---|---|---|---|---|
| Enclosed by barrier islands | - - | ++ | - - | - - |
| Surrounded by land on 3 sides; outlet to larger body of water may contain a sill | ++ | - - | - - | - - |
| Surrounded by 2 bodies of land with 2 outlets to water bodies on the others | - - | - | - - | ++ |
| Relatively distant from land, > 20 km | - - | - - | + | + |
| Receives glacial freshwater input | + | - - | - - | - |
| Receives riverine (meteoric water) input | - | ++ | + | - - |
| Seasonal salinity fluctuations of < 25 | - - | ++ | - | - - |
| Macroalgal food web end-member available | + | - | - | + |

Coastscape characteristics are denoted as ++ (almost always with few exceptions), + (oftentimes yes), - (oftentimes no), and - - (rarely with few exceptions).

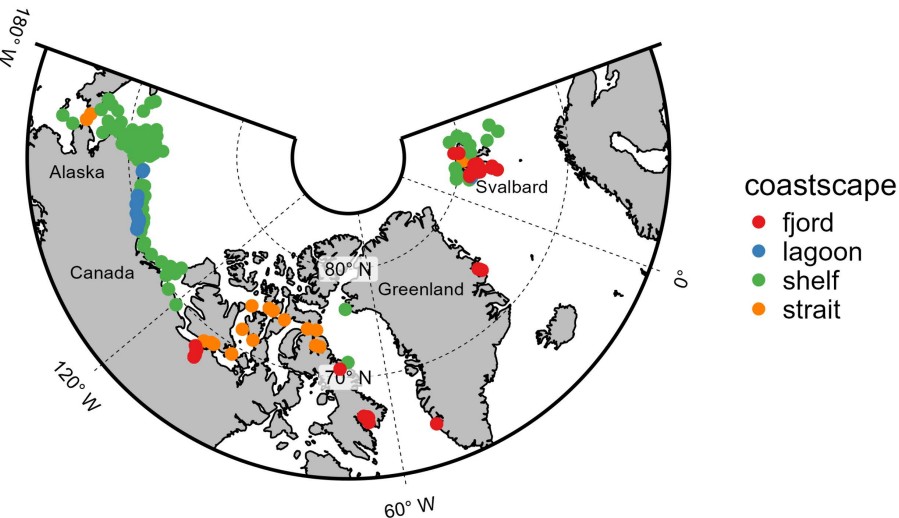

**Fig 1. Sampling locations across the Arctic where data were aggregated.** Colors denote coastscape. Each point may represent many data rows.

in addition to the coastscapes, *a priori* regional designations were assigned to data based on collection site longitude. The Northern Bering/Chukchi Seas sector was defined as sites west of 157 °W, the Beaufort Sea sector was between 123°W and 157°W, the Canadian Arctic Archipelago sector was between 78°W and 123°W, the Baffin Bay sector was between 45°W and 78°W, the East Greenland sector was between 0° and 45°W, and the Svalbard sector was east of 0°.

## Bathymetry requirements

Sample collection depth was considered when consolidating the dataset since it has a paradoxical relationship to the distance to coastal environments in some instances. For example, depths can be > 300 m in some fjord and strait environments that are in close proximity to a coastline, whereas the broad, shallow shelves of the Chukchi and Northern Bering Seas host locations that are > 100 km from the nearest coast yet are < 100 m deep [1]. Instead of using a single bathymetric requirement across the entire dataset, collection sites deeper than 45 m in the Northern Bering, Chukchi, and Beaufort Sea shelves were omitted from analysis. Some shelf sites around Svalbard and the Canadian Archipelago were deeper than this but were retained in the dataset due to their close proximity to their respective coastlines. These restrictions distilled the dataset to 8,803 rows.

## Taxonomic classification, feeding habit assignment, and end-member synonymization

Food web end-members were collated from a 20-y period from 2 April 2002–5 August 2022 and totaled 1,078 rows. 86% of end-member data were classified as either pelagic particulate organic matter (pPOM) or sediment particulate organic matter (sPOM), while the remaining end-members consisted of sea-ice particulate organic matter (iPOM), benthic microalgae, red algae, and brown algae. Ice algae were grouped with iPOM. Generically described particulate organic matter and all phytoplankton collected from all depths were grouped as pPOM. Sediment or sediment organic matter was grouped into sPOM; when specified, sPOM was collected from the most surficial 0.5–2 cm of undisturbed surface sediment. All macroalgal species were grouped as macroalgae. Only pPOM and sPOM were tested for temporal changes since they were consistently represented throughout the collection period.

The remaining 7,725 rows of data were for consumers that consisted of 639 distinct taxa from 339 genera and 15 phyla collected between 6 May 1999 and 17 August 2022. Taxonomic classifications were attributed from the World Register of Marine Species (www.marinespecies.org). Feeding habits were assigned from either The Arctic Traits Database [68], Macdonald et al. [69], or Ehrman et al. [50]. If species-level information was not available, information from congeners or cofamilials was used. Feeding habits were aggregated into four distinct groups based on similar ecological function according to McTigue & Dunton [57]. The categorizations allowed for subsequent analysis across Arctic ecosystems at a functional level without the caveats of biogeographical species distributions limiting comparisons since few cosmopolitan species existed with sufficient replication within the dataset.

Grazers, surface deposit feeders, and subsurface deposit feeders were combined as deposit feeders. Filter feeders were combined with suspension feeders. Parasites were combined with predators. Less than 10% of data were vertebrates consisting of fish and mammals, all of which were classified into the predator feeding habit. Scavengers and omnivores were combined into the opportunist/scavengers group. The dataset was dominated by benthic taxa but included some pelagic and sympagic species. The only zooplanktonic organisms included in analysis were Copepoda, *Balanus* sp., and Pycnogonida. Bulk zooplankton and microzooplankton were not used.

## Lipid correction of stable isotope data

Lipids are generally $^{13}$C-depleted compared to other tissues and could potentially influence the comparison of stable carbon isotope values of organisms of varying lipid content [70]. Global mathematical corrections exist [71] but consist of broad categorizations at the level of marine fishes or marine invertebrates, for example, and the organisms used to determine the correction factors were not Arctic fauna.

A subset of aggregated data possessed stable carbon and nitrogen isotope values on paired lipid-extracted and non-extracted tissue samples from a variety of Arctic invertebrate fauna. Briefly, for select samples across a large number of invertebrate taxa and feeding types collected by the Arctic Marine Biodiversity Observation Network (AMBON) from the Chukchi Sea shelf, samples were divided into two parts. Lipids were chemically extracted from one part of each of these samples while the other remained untreated. The sub-samples selected for lipid extraction were treated at least three times in 2:1 chloroform:methanol (v:v) for at least 12 h for each extraction. After the final extraction, samples were dried and measured for stable isotope composition in addition to the non-extracted partner sample.

Comparisons among lipid-extracted and non-extracted data were made with the goal of creating taxon-specific mathematical stable isotope value corrections for lipid content. Although taxa were classified at least to the genus level within the raw data, taxa were aggregated at the phylum level to increase sample size for linear regression analysis. Wilcoxon sign rank tests (non-parametric paired t-test) were used to determine if there was a significant difference between lipid-extracted and non-extracted tissue samples. The exception to this procedure was for the copepod genus *Calanus* that, as a pelagic primary consumer, was analyzed separately from other typically benthic Arthropoda. We used a direct comparison of stable isotope values between paired extracted vs. non-extracted tissues to determine the linear regression correction. Previous work has demonstrated that C:N values can be indicators of lipid content and used to apply mathematical corrections (e.g., Post et al. [70]). We observed small ranges of C:N values despite large ranges of stable isotope values in the test dataset and opted not to incorporate C:N values into the mathematical correction. Moreover, C:N values were not available for all data collated for this analysis.

For calcifying organisms (e.g., Echinodermata), stable isotope values were compared between acidified tissue and lipid-extracted tissue after acidification. Significance level alpha was set at 0.05 and a Bonferroni adjustment was applied for multiple comparisons (Table 2). As a conservative approach, linear regression equations derived from this comparative sample set were used to correct non-lipid extracted samples at the genus level in the rest of the aggregated dataset instead of at higher taxonomic levels. Overall, 13% of data consisting of 53 genera received a mathematical correction using linear regression equations (Tables 2 and S2). On average, after mathematical correction, $\delta^{13}$C and $\delta^{15}$N values changed by +1.4 ± 1.2‰ and +0.3 ± 0.6‰, respectively, with median changes of +1.1‰ and 0‰.

**Table 2. Mathematical treatments applied to data to correct stable isotope values for lipid content.**

| Isotope ratio | Treatment | Taxa | Equation |
|---|---|---|---|
| $\delta^{13}$C | HCl + lipid extraction | Arthropoda | y = 0.76x − 3.9 |
| $\delta^{15}$N | lipid extraction | Arthropoda | y = 1.02x + 0.04 |
| $\delta^{13}$C | HCl + lipid extraction | Bryozoa | y = 0.92x − 1.3 |
| $\delta^{15}$N | HCl + lipid extraction | Bryozoa | y = 0.96x + 0.08 |
| $\delta^{13}$C | lipid extraction | *Calanus* | y = 1.2x + 7.6 |
| $\delta^{13}$C | lipid extraction | Chordata | y = 0.74x − 4.4 |
| $\delta^{15}$N | lipid extraction | Chordata | y = 0.66x + 5.0 |
| $\delta^{13}$C | HCl + lipid extraction | Cnidaria | y = 1.07x + 2.5 |
| $\delta^{13}$C | HCl + lipid extraction | Echinodermata | y = 0.82x − 2.3 |
| $\delta^{15}$N | lipid extraction | Echinodermata | y = 1.21x − 0.91 |
| $\delta^{13}$C | lipid extraction | Mollusca | y = 0.89x − 1.7 |
| $\delta^{13}$C | lipid extraction | Porifera | y = 0.86x − 2.0 |
| $\delta^{15}$N | lipid extraction | Porifera | y = 1.36x − 1.79 |

Only significant relationships are shown. For the 'lipid extraction' treatment, paired samples were compared between lipid extraction and no treatment. For the 'HCl + lipid extraction" treatment, paired samples were compared between acidified samples that were subsequently lipid extracted versus samples only acidified. Alpha was 0.05 with Bonferroni adjustments made for multiple tests. Note that *Calanus* was tested outside of Arthropoda. Genera mathematically corrected within each Phylum are listed in S2 Table.

## Statistical applications

To test the first hypothesis, linear regression analysis was applied to detect temporal changes in stable isotope values. In most cases, data were not normally distributed. While this does not impact linear regression, non-parametric tests were used in subsequent post-hoc tests. We used the non-parametric correlation coefficient Kendall's tau ($\tau$). Instead of using a one-way ANOVA, a non-parametric Kruskal-Wallis test with a Dunn Test post-hoc pairwise comparison was used. Alpha ($\alpha$) was set to 0.001 to be conservative and reduce the likelihood of false positives, unless otherwise noted. All analyses and data visualization were performed using R v4.2.2 [72] and RStudio [73]. The map was created using the 'grfxtools' package in R, which uses Natural Earth basemap data from the public domain [74].

## Isotopic niche regions ($N_R$) and niche overlap

To test the second and third hypotheses, we measured isotopic niches using the approach and tools described by Lysy et al. [75] and Swanson et al. [30] with the R package *nicheROVER* to quantify isotopic niche regions ($N_R$) of different groups. As mathematically described by Swanson et al. [30], the method uses a Bayesian framework to estimate the n-dimensional space ($N_R$) that a data point has an $\alpha$ probability of being found, where n is the number of isotopes used and $\alpha$ is the user-defined probability, in this case 95%. Then probabilistic niche overlap between two groups is conducted by estimating the probability that a data point randomly selected from one group would fall within the $N_R$ space of the other group using a Monte Carlo technique. $N_R$ was informed from bivariate $\delta^{13}$C and $\delta^{15}$N values. The method is robust to differences in sample size, performs well with large sample sizes, and, unlike geometric methods, accounts for species-specific distributions in bivariate space when estimating the probability of overlap [30]. Uncertainty was accounted for by using a Bayesian approach with a noninformative normal-inverse-Wishart prior distribution and 10,000 random permutations to calculate the posterior mean and 95% credible intervals of probabilities of niche overlap. We apply the isotopic niche as a proxy for a trophic niche but recognize they are not synonymous. Both terms are used here with isotopic niche pertaining to stable isotope values, calculated $N_R$, and resulting statistical comparison, whereas trophic niche refers to resource use within the ecosystem.

## Investigating potential effects of depth on $\delta^{15}$N

A positive relationship between $\delta^{15}$N and depth has been documented for benthic deposit and suspension feeders, attributed to biogeochemical alterations to POM particles as they sink through the water column [76,77]. Accordingly, relationships between suspension feeder $\delta^{15}$N and depth were investigated using linear regressions for each coastscape independently to ensure results were not affected by variability in depth range among coastscapes. Depth was $\log_{10}$-transformed when it improved normality of residuals (shelf and strait coastscapes), and Spearman's rank correlation analysis was used when assumptions of linear regression were not met (fjord coastscape).

# Results

## Isotopic and temporal patterns of end-members

$\delta^{13}$C and $\delta^{15}$N values for pPOM and sPOM exhibited overlap among coastscapes, although there were significant differences among groups (Fig 2). For example, lagoon pPOM and sPOM mostly exhibited $^{13}$C-depleted values (less than −25‰) compared to other coastscapes. The presence of low $\delta^{15}$N values (−5.8 to 11.6‰) of fjord pPOM contributed to their significant difference to all other coastscapes (Table 3). Fjords also exhibited the widest range of $\delta^{13}$C values for pPOM (−33.5 to −20.8‰) and the most $^{13}$C-enriched sPOM value at −19.0‰ (Table 3). Straits exhibited a wide range of $\delta^{13}$C values for pPOM that overlapped with other coastscapes, while SPOM from this coastscape was constrained to a narrow range of relatively $^{13}$C-enriched values (−24.4 to −22.2‰).

The linear regressions between collection date and pPOM $\delta^{13}$C values ($p < 0.001$, Kendall's $\tau = −0.267$) and collection date and sPOM $\delta^{13}$C values ($p < 0.001$, Kendall's $\tau = −0.446$) revealed significant negative trends with slopes indicating

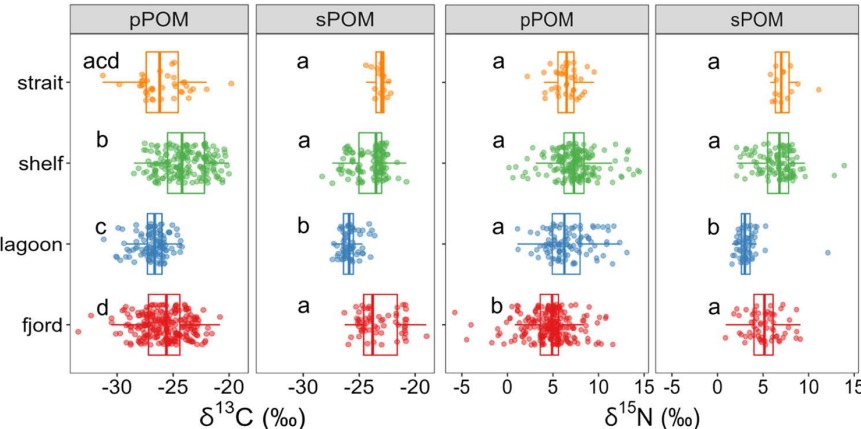

**Fig 2. Distribution of stable carbon and nitrogen values for pPOM and sPOM.** Data points were plotted over boxplots that summarize distributions. Letters denote significant differences within each column. A Dunn test was used for post-hoc multiple comparisons, and alpha was modified with Bonferonni correction.

decadal decreases of −2.1 and −2.2 ‰, respectively (Fig 3; Table 4). Likewise, the trend between collection date and sPOM δ15N values (p < 0.001, Kendall's τ = −0.478) was significantly negative with a decadal change of −3.1 ‰ per decade (Table 4). The linear regression between collection date and pPOM δ15N was not significant. To evaluate if these negative trends were influenced by non-uniform coastscape occurrences, specifically lagoons since they bookend the dataset, we excluded the lagoon samples and re-ran the regression analysis. The significant temporal trends described above remained intact and within the same magnitude when lagoon data were excluded (δ13C for pPOM: y = −0.000619x − 22.8; δ13C for sPOM: y = −0.000618–23.0) versus included (δ13C for pPOM: y = −0.000567x − 23.0; δ13C for sPOM: y = −0.000613x − 21.9; see Table 4).

## Isotopic and temporal patterns of consumers

Overall δ13C values ranged from −30.2 to −11.5‰, and δ15N values ranged from 0.3 to 21.3‰. The isotopic centroid for the entire dataset was −20.4 and 11.6‰ for δ13C and δ15N, respectively. While stable isotope value ranges overlapped among all feeding groups, statistical differences existed between some groups. For δ13C in all feeding modes, there were no significant differences between straits and shelves, or between lagoons and fjords (Fig 4). Median δ13C values for consumers from lagoons and fjords were significantly different and more 13C-depleted than those in shelves or straits.

Patterns in δ15N values for consumers were similar to those in δ13C, except for the opportunist/scavenger group. Overall, δ15N values were overlapping among coastscapes for each feeding habit, although significant differences were detectable among groups (Fig 4). Median values of lagoon and fjord consumers were generally more 15N-depleted than straits and shelves except for opportunist/scavengers from straits, which were also 15N-depleted compared to the shelf counterparts (Fig 4).

Over the period represented by this dataset (~ 2 decades), the linear regression between collection date and all consumer δ13C values was significant and negative (Table 4; S1 Fig). Specifically, the suspension feeder, deposit feeder, and opportunist/scavenger groups exhibited negative linear regressions between collection date and δ13C values, with slopes that extrapolated to respective changes of −1.1‰, −1.1‰, and −0.9‰ per decade (Table 4). The linear regression between δ15N and time was not significant when using all consumers in the dataset. Only the opportunist/scavenger feeding habit showed a significant linear regression between collection date and δ15N values.

**Table 3. Summary statistics of δ¹³C and δ¹⁵N for end-members and consumer feeding habits.**

| Group | coastscape | n | δ¹³C (‰) | | | | δ¹⁵N (‰) | | | |
|---|---|---|---|---|---|---|---|---|---|---|
| | | | Minimum | Maximum | Median | Mean±s.d. | Minimum | Maximum | Median | Mean±s.d. |
| pPOM | fjord | 237 | −33.5 | −20.8 | −25.6 | −25.7±2.0 | −5.8 | 11.6 | 4.9 | 4.6±2.4 |
| | lagoon | 155 | −31.5 | −21.7 | −26.8 | −26.7±1.5 | 1.1 | 13.1 | 6.5 | 6.7±2.2 |
| | shelf | 197 | −28.5 | −20.0 | −24.3 | −24.1±2.0 | 0.2 | 14.5 | 7.3 | 7.1±2.4 |
| | strait | 39 | −31.3 | −19.8 | −26.2 | −25.9±2.1 | 2.2 | 9.5 | 6.5 | 6.4±1.4 |
| sPOM | fjord | 62 | −26.3 | −19.0 | −23.8 | −23.2±1.7 | 0.9 | 9.1 | 5.2 | 5.1±1.7 |
| | lagoon | 101 | −28.3 | −23.1 | −26.0 | −25.8±0.9 | 0.1 | 12.1 | 3.1 | 3.2±1.2 |
| | shelf | 126 | −28.3 | −20.8 | −23.6 | −24.2±1.5 | 1.5 | 13.9 | 6.6 | 6.0±2.2 |
| | strait | 15 | −24.4 | −22.2 | −23.0 | −23.1±0.5 | 5.8 | 11.1 | 7.0 | 7.2±1.3 |
| iPOM | fjord | 11 | −30.2 | −14.4 | −16.7 | −19.3±6.2 | 0.2 | 5.1 | 4.8 | 4.0±1.5 |
| | lagoon | 11 | −28.4 | −25.0 | −26.2 | −26.3±1.0 | 5.1 | 8.6 | 7.0 | 6.7±1.0 |
| | shelf | 44 | −37.0 | −5.5 | −21.6 | −18.6±8.6 | 3.0 | 12.5 | 7.9 | 7.9±2.1 |
| macroalgae | fjord | 57 | −37.5 | −14.7 | −22.7 | −25.2±7.0 | 0.7 | 13.8 | 4.3 | 4.4±2.2 |
| | lagoon | 7 | −25.1 | −21.8 | −24.3 | −23.9±1.1 | 2.7 | 7.2 | 5.2 | 5.1±1.4 |
| | shelf | 4 | −36.9 | −21.0 | −22.5 | −25.7±7.4 | 5.6 | 11.4 | 7.3 | 7.9±2.7 |
| | strait | 12 | −27.8 | −18.4 | −24.6 | −24.0±3.0 | 5.0 | 14.2 | 11.5 | 10.3±3.4 |
| suspension feeder | fjord | 629 | −30.2 | −14.5 | −21.9 | −22.1±2.1 | 3.9 | 18.6 | 7.8 | 8.0±2.0 |
| | lagoon | 252 | −27.9 | −13.3 | −21.9 | −21.9±2.2 | 4.1 | 14.7 | 9.8 | 9.8±1.9 |
| | shelf | 952 | −28.1 | −13.0 | −20.6 | −20.6±2.0 | 3.2 | 18.2 | 10.6 | 10.5±2.1 |
| | strait | 154 | −26.1 | −14.5 | −20.5 | −20.4±2.6 | 5.8 | 19.9 | 11.3 | 11.9±3.1 |
| deposit feeder | fjord | 400 | −26.4 | −15.4 | −20.9 | −20.8±1.7 | 2.1 | 16.3 | 8.7 | 8.9±2.4 |
| | lagoon | 164 | −29.3 | −15.2 | −20.4 | −20.2±2.2 | 6.5 | 14.0 | 9.5 | 9.7±1.5 |
| | shelf | 638 | −26.7 | −13.1 | −19.3 | −19.6±1.9 | 3.4 | 19.4 | 10.6 | 10.9±2.5 |
| | strait | 58 | −24.0 | −14.4 | −18.9 | −19.0±2.1 | 6.3 | 15.5 | 12.2 | 11.7±2.4 |
| opportunist/scavenger | fjord | 289 | −28.4 | −14.6 | −20.1 | −20.5±2.4 | 5.1 | 17.1 | 10.3 | 10.3±1.9 |
| | lagoon | 97 | −24.4 | −14.6 | −20.4 | −20.4±2.0 | 1.1 | 16.6 | 10.4 | 10.4±2.8 |
| | shelf | 853 | −25.8 | −11.5 | −18.8 | −19.2±1.7 | 0.3 | 19.2 | 14.1 | 13.5±2.1 |
| | strait | 94 | −26.3 | −13.5 | −19.0 | −19.0±2.6 | 5.4 | 16.3 | 10.6 | 10.6±2.7 |
| predator | fjord | 864 | −27.8 | −12.9 | −20.7 | −20.9±2.5 | 5.5 | 18.4 | 11.0 | 11.1±2.4 |
| | lagoon | 318 | −28.2 | −14.4 | −20.6 | −20.6±2.5 | 3.7 | 21.3 | 12.4 | 12.3±2.2 |
| | shelf | 1822 | −28.0 | −12.4 | −19.9 | −20.0±2.6 | 5.0 | 21.2 | 14.2 | 13.8±2.0 |
| | strait | 141 | −27.5 | −12.4 | −19.5 | −19.6±2.6 | 6.3 | 19.4 | 13.8 | 13.2±3.1 |

s.d. = standard deviation.

Mean posterior estimates of niche overlap were generally high across coastscapes (Table 5; S2–S5 Figs), mostly resulting in an estimated >70% niche region space ($N_R$) overlap. As an exception to the trend, opportunist/scavengers from shelves exhibited ~50% overlap with other coastscapes. This coincides with significantly higher δ¹⁵N values for opportunist/scavengers in shelves than those in fjords, straits, or lagoons (Fig 4).

## Comparison of fjord and shelf isotopic niches across sectors

The deposit and suspension feeders sampled in fjord coastscapes exhibited greater among-sector variability along the δ¹⁵N axis than along the δ¹³C axis (Fig 5). Notably, there was a high probability of niche overlap (> 96%) estimated for East Greenland consumers in both the Canadian Arctic Archipelago and Svalbard isotopic niches (Table 6). Consumers

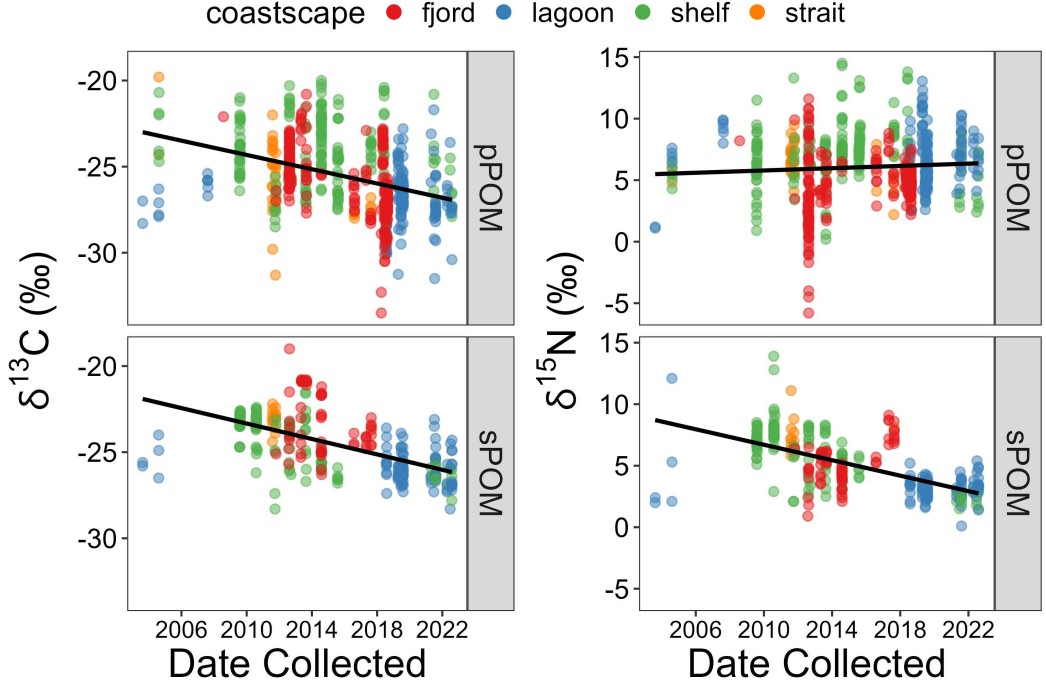

**Fig 3. Linear regressions of pPOM and sPOM stable carbon and nitrogen isotope values versus collection date.** Color represents the coastscape from which each data point was sampled. See Table 4 for regression statistics. The significant trend between collection date and pPOM $\delta^{13}C$ (p < 0.001, Kendall's $\tau = -0.267$) and collection date and sPOM $\delta^{13}C$ (p < 0.001, Kendall's $\tau = -0.446$) translate to changes of −2.1 and −2.2 ‰ per decade, respectively, for coastal Arctic end-members. The significant trend between collection date and sPOM $\delta^{15}N$ (p < 0.001, Kendall's $\tau = -0.478$) translates to a change of −3.1 ‰ per decade.

sampled in the Canadian Arctic Archipelago had the largest $\delta^{15}N$ and $\delta^{13}C$ ranges (Fig 5; Table 6); thus, the probabilities of niche overlap of consumers from Svalbard or East Greenland falling into the isotopic niche of Canadian Archipelago consumers was relatively high (Table 6). Conversely, Canadian Archipelago consumers had a lower probability of falling into the isotopic niches of the other two sectors that exhibited smaller $N_R$.

In contrast, the deposit and suspension feeders sampled in shelf coastscapes exhibited among-sector variability along the $\delta^{13}C$ and $\delta^{15}N$ axes (Fig 6). However, the Beaufort Sea and Northern Bering/Chukchi Seas shelf consumers exhibited a higher probability of overlap than either did with shelf consumers from the Svalbard shelf (Table 7), attributable mostly to lower mean $\delta^{15}N$ of the Svalbard shelf niche.

### Investigating potential effects of depth on $\delta^{15}N$

Linear relationships between benthic deposit and suspension feeder $\delta^{15}N$ and depth were not significant for lagoons, fjords, or shelves. There was a significant, but weak positive correlation between deposit and suspension feeder $\delta^{15}N$ and $\log_{10}$ depth for straits (depth range 17–789 m; Spearman's correlation, *rho* = 0.17, *p* < 0.001).

### Discussion

### Temporal changes in stable isotope values

Contrary to the null hypothesis that stable isotope values of end-members would not change over a decadal scale, pPOM and sPOM became 2.1‰ and 2.2‰ more [13]C-depleted per decade, respectively, from 2002–2022 across Arctic coastal

**Table 4. Results from linear regression analysis for δ¹³C or δ¹⁵N and collection date.**

| Group | Isotope ratio | Intercept | Slope | Slope p-value | Kendall's τ | Decadal Δ (‰) |
|---|---|---|---|---|---|---|
| **Endmembers** | **δ¹³C** | −21.9 | **−0.000623** | **<0.001** | −0.263 | **−2.3** |
| **Endmembers** | **δ¹⁵N** | 6.8 | **−0.000212** | **<0.001** | −0.116 | **−0.8** |
| **pPOM** | **δ¹³C** | −23.0 | **−0.000567** | **<0.001** | −0.267 | **−2.1** |
| pPOM | δ¹⁵N | 5.5 | 0.000127 | 0.077 | 0.009 | |
| **sPOM** | **δ¹³C** | −21.9 | **−0.000613** | **<0.001** | −0.446 | **−2.2** |
| **sPOM** | **δ¹⁵N** | 8.7 | **−0.000861** | **<0.001** | −0.478 | **−3.1** |
| **All consumers** | **δ¹³C** | −19.5 | **−0.000168** | **<0.001** | −0.063 | **−0.6** |
| All consumers | δ¹⁵N | 11.6 | 0.000002 | 0.910 | −0.023 | |
| **Suspension feeders** | **δ¹³C** | −19.6 | **−0.000295** | **<0.001** | −0.183 | **−1.1** |
| Suspension feeders | δ¹⁵N | 9.2 | 0.000099 | 0.004 | 0.0197 | |
| **Deposit feeders** | **δ¹³C** | −18.4 | **−0.000307** | **<0.001** | −0.168 | **−1.1** |
| Deposit feeders | δ¹⁵N | 10.7 | −0.000089 | 0.059 | −0.070 | |
| **Opportunist/scavenger** | **δ¹³C** | −18.2 | **−0.000235** | **<0.001** | −0.065 | **−0.9** |
| **Opportunist/scavenger** | **δ¹⁵N** | 10.9 | **0.000267** | **<0.001** | 0.0759 | **+1.0** |
| Predator | δ¹³C | −20.2 | 0.000019 | 0.514 | 0.0414 | |
| Predator | δ¹⁵N | 12.6 | 0.000061 | 0.032 | 0.0170 | |

For regression analysis to detect change over time, collection dates were represented as Julian dates with the origin as the earliest sample collected. A Bonferroni adjustment (α/n) was used to conservatively detect significance across 16 tests (i.e., 0.001/16); original p-values are displayed on the table. The non-parametric correlation coefficient Kendall's τ was used, which is akin to Pearson's r. Significant slopes according to the Bonferroni adjustment are bolded. Decadal change (Δ) was calculated by multiplying the slope, which was determined using the time unit day, by the number of days in 10 years, or 3650. Only significant Decadal Δ are shown.

environments. This aligns with results from de la Vega et al. [35] who found a ¹³C-depletion of ~1.5‰ per decade for pPOM across the Arctic basin and adjacent shelf seas. Those authors attributed the trend, in part, to the Suess effect, which ascribes the decreasing δ¹³C value of atmospheric $CO_2$ to the emissions from ¹³C-depleted fossil fuels [78]. The atmospheric $CO_2$ signature becomes reflected in the δ¹³C value of dissolved inorganic carbon (DIC) in marine waters and, subsequently, by primary producers after carbon fixation [79]. With their pan-Arctic assessment, de la Vega et al. [35] showed DIC became ~0.1‰ more ¹³C-depleted per decade between 1977–2014 due to the Suess effect. However, that rate of change for DIC was an order of magnitude less than the ~2‰ decadal change in pPOM and sPOM found by this study; therefore, the Suess effect can only partially explain the change in the Arctic primary producers observed in this study.

As suggested by de la Vega et al. [35], one explanation for these end-members becoming more ¹³C-depleted over time is a change in the relative composition of pPOM and sPOM, since they are only operationally defined and consist of a mixture of organic matter of varied origin. A decreased contribution of sea ice algae [35], which are typically ¹³C-enriched compared to phytoplankton [80], increased contributions of ¹³C-depleted terrestrially-derived organic matter [15], or both, would account for the observed isotopic changes in pPOM and sPOM. Greater terrestrial inputs to coastal pPOM and sPOM are plausible given the increase in river discharge, thawing permafrost, and coastal erosion [5,7,8]. Freshwater aquatic microalgae, with δ¹³C values of −33.1 ± 4.7‰, constitute 39–60% of the POM exported by rivers to coastal environments [18]. Increased freshwater export of this ¹³C-depleted end-member could also contribute to the decreasing pPOM and sPOM values in coastal Arctic ecosystems. It is also possible that increased inputs of riverine DIC and/or DIC remineralized from terrestrial organic matter in coastal habitats would cause ¹³C-depletion of the DIC pool [81]. With the baseline stable carbon isotope value for primary producers left-shifted, the autochthonous microalgal components of the pPOM and sPOM pools would become ¹³C-depleted over time as well.

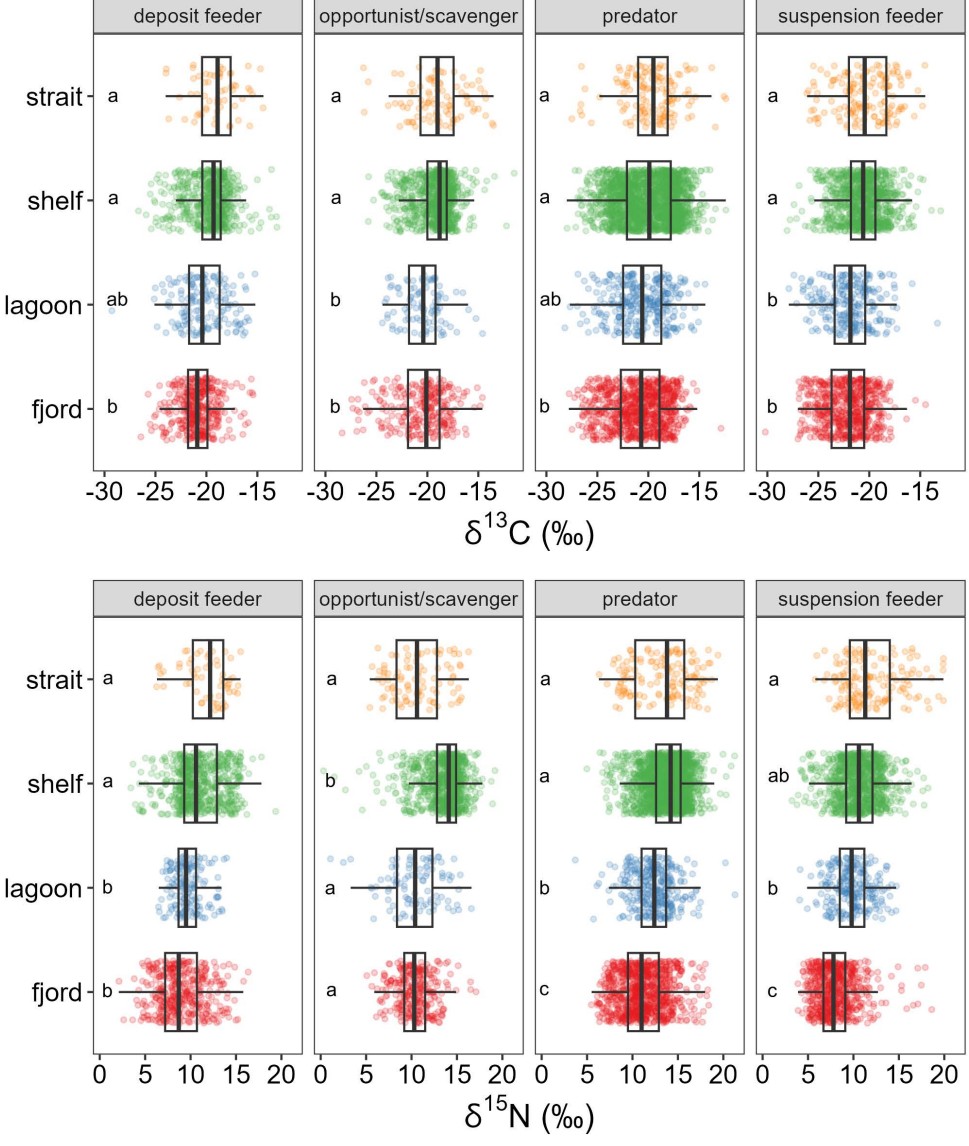

**Fig 4. Distribution of stable isotope values for consumers faceted by coastscape and feeding habit.** Significant differences denoted with letters within each facet. A Dunn test was used for post-hoc multiple comparisons, and alpha was modified with Bonferonni correction. Note different x-axes.

The similar temporal trends of $\delta^{13}$C exhibited by the pPOM and sPOM pools appear to support the tight coupling between the two pools often observed in the Arctic [64,65,82–86]. Therefore, it was unexpected that the $\delta^{15}$N values for the two pools would exhibit different temporal trends (Fig 3). One explanation for no temporal trend for $\delta^{15}$N in pPOM might stem from the timeframe over which it forms and the inorganic nitrogen sources it assimilates compared to the sPOM pool. As the living microalgal component of pPOM blooms during the short Arctic growing season, it would assimilate and reflect the $\delta^{15}$N signature of its inorganic nitrogen pool, typically marine nitrate. Coastal Arctic marine nitrate $\delta^{15}$N values typically range between 5–8‰ [87], and the mean and median values from pPOM in this dataset fall within that range at 6.1 and 6.0‰, respectively. Therefore, we may be observing predominant usage of marine nitrate by coastal primary producers in the pPOM pool over the past two decades, despite recent findings that terrestrial nitrogen fuels

**Table 5. Quantification of isotopic niche overlap for feeding habits between coastscapes.**

|  | fjord | lagoon | shelf | strait |
|---|---|---|---|---|
| **Predator** |  |  |  |  |
| fjord | NA | 93.3 (90.2, 95.9) | 72.4 (67.0, 75.9) | 94.5 (90.9, 97.4) |
| lagoon | 86.4 (82.1, 90.1) | NA | 81.2 (76.6, 85.4) | 91.5 (86.6, 95.7) |
| shelf | 80.4 (76.7, 83.8) | 92.2 (88.5, 95.3) | NA | 94.0 (90.4, 96.9) |
| strait | 80.6 (74.2, 85.9) | 86.3 (80.0, 91.7) | 80.5 (74.6, 85.7) | NA |
| **Opportunist/scavenger** |  |  |  |  |
| fjord | NA | 91.3 (83.8, 97.1) | 55.6 (49.6, 61.5) | 95.6 (91.1, 98.5) |
| lagoon | 76.2 (65.1, 85.7) | NA | 48.7 (37.4, 59.6) | 84.27 (73.0, 93.4) |
| shelf | 70.1 (62.1, 78.0) | 94.3 (84.3, 99.2) | NA | 94.2 (87.6, 98.5) |
| strait | 80.5 (72.4, 87.6) | 82.3 (71.3, 92.3) | 51.9 (43.9, 60.2) | NA |
| **Deposit feeder** |  |  |  |  |
| fjord | NA | 77.5 (69.2, 85.5) | 87.4 (83.4, 91.1) | 83.3 (71.4, 93.0) |
| lagoon | 91.9 (85.8, 96.2) | NA | 91.9 (85.9, 96.2) | 92.5 (84.0, 98.0) |
| shelf | 87.9 (83.8, 91.5) | 75.0 (66.6, 83.4) | NA | 94.4 (89.4, 98.0) |
| strait | 73.7 (62.8, 83.1) | 62.6 (49.9, 75.6) | 88.3 (80.5, 94.1) | NA |
| **Suspension feeder** |  |  |  |  |
| fjord | NA | 93.1 (88.6, 96.6) | 73.4 (68.7, 78.0) | 89.2 (82.8, 95.0) |
| lagoon | 89.4 (84.2, 93.6) | NA | 79.3 (72.7, 85.3) | 92.3 (86.2, 96.7) |
| shelf | 79.1 (74.4, 83.7) | 91.8 (86.3, 96.2) | NA | 98.4 (96.9, 99.5) |
| strait | 53.9 (46.7, 60.8) | 68.0 (58.3, 77.4) | 74.5 (68.1, 80.1) | NA |

Values are mean posterior estimates of niche region ($N_R$) overlap probability. Values in parentheses are the 95% credible interval (CI) range (2.5%, 97.5%) for consumers in each feeding habit between coastscapes. Matrix should be read as the probability that an individual from the coastscape indicated in the row will be found within the isotopic niche indicated in the column.

about one-third of Arctic primary production [17]. Conversely, sPOM exhibited the trend of $^{15}$N-depletion over time, suggesting over the two decades analyzed here its association with marine nitrate weakened while its association with terrestrial nitrogen sources strengthened. Alternatively, the sPOM $\delta^{15}$N trend might not be related to microalgal assimilation of nitrate, but rather a larger proportion of $^{15}$N-depleted allochthonous terrestrial OM entering the sPOM pool that is not retained in pPOM. This seems plausible since coastal sediments are repositories of organic matter [88] and accumulate over long periods of time compared to the water column, which constantly refreshes with new microalgal growth. Interestingly, terrestrial OM can remain suspended compared to marine particles that sink faster and contribute more to sPOM over millennial timescales [89], so terrestrial OM deposition near the coast cannot be broadly assumed, especially in deeper zones. Still, biomarker evidence demonstrates that high proportions of sPOM in the coastal Arctic are derived from terrestrial sources [90,91]. Furthermore, it has been noted in lagoons [15], shelves [32,48,58], and fjords [45] that sPOM was consistently 0.4 to 4.0‰ more $^{15}$N-depleted than pPOM.

Consumers of pPOM and sPOM also reflected a $^{13}$C-depletion trend over time. While a significant trend was found with all pooled consumers, this was driven by suspension feeders, deposit feeders, and opportunist/scavengers (Table 4). The regression slope for those three groups extrapolated to ~ −1‰ per decade for $\delta^{13}$C, which interestingly was less change than what was detected in the end-members. A similar pattern was also detected by de la Vega et al. [35], who measured significant $^{13}$C-depletion for marine mammals over decadal timescales but at a lesser rate than pPOM was changing. This may indicate that benthic invertebrates preferentially select for organic matter sources with more nutritional quality [92]. Thus, increasing inputs of terrestrial organic matter in end-member pools are not reflected to the same extent in the diet of benthic invertebrates because of the low nutritional quality of terrestrial organic matter.

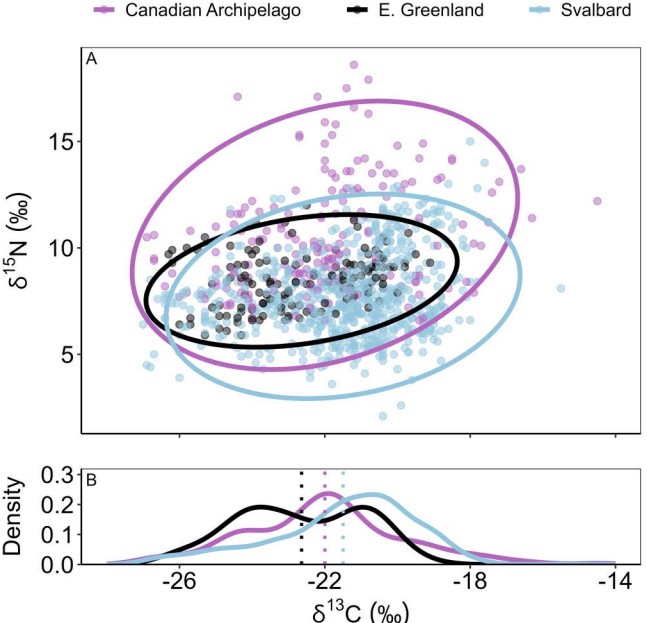

**Fig 5. Niche overlap within fjord communities.** (A) Mean posterior estimates of isotopic niche regions constructed from $\delta^{15}N$ and $\delta^{13}C$ values for benthic deposit and suspension feeders sampled in fjord coastscapes, in three geographic sectors. Niche regions were calculated at alpha = 0.95. (B) Density distribution of $\delta^{13}C$ data from each sector, with dashed lines indicating the sample mean.

**Table 6. Quantification of isotopic niche overlap in fjords.**

| FJORD | Canadian Archipelago | E. Greenland | Svalbard |
|---|---|---|---|
| **Canadian Archipelago** | NA | 52.6 (44.8, 60.9) | 66.4 (59.8, 73.0) |
| **E. Greenland** | 98.7 (96.7, 99.7) | NA | 96.8 (94.0, 98.6) |
| **Svalbard** | 87.4 (80.7, 93.0) | 70.5 (63.2, 77.6) | NA |

Mean posterior estimates of overlap probability and 95% credible interval (CI) range for suspension and deposit feeders from each sector within fjords. Numbers represent the probability that an individual from the sector indicated in the row will be found within the niche indicated in the column. Values are reported as mean posterior estimate with lower and upper limits in parentheses (2.5% CI, 97.5% CI).

As a caveat, we acknowledge no single end-member or consumer group is necessarily changing by the rates indicated by the regression slopes of pooled samples. These trends detected the overall change of aggregated data across the entire Arctic coastal areas as our intent was to examine coarse, pan-Arctic patterns.

## Isotopic niche overlap

The second hypothesis that organic matter sources would be differentially assimilated by Arctic consumers based on geomorphological differences between coastscapes was not supported. Consumers of the same feeding habit between coastscapes exhibited high isotopic niche overlap (Table 5). Previous examples have shown that different feeding habits can share trophic niches within specific Arctic coastscapes, for example, within Arctic lagoon systems [15,93], on the Pacific Arctic shelf [32,94], in strait regions of the Canadian Arctic Archipelago [95,96], and in Arctic fjords [60,66]. Most commonly, this high trophic niche overlap is attributed to the high trophic plasticity of many Arctic invertebrate consumers that allows them to capitalize on a wide variety of food sources, a strategy that is considered adaptive due to the high

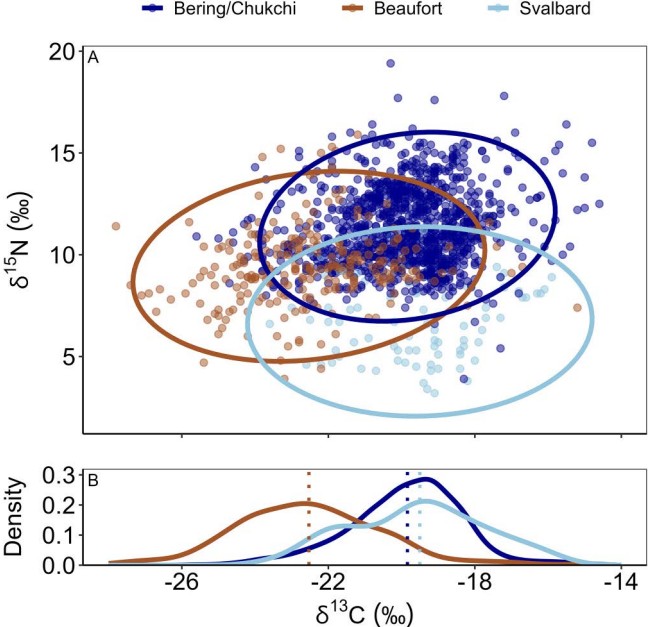

**Fig 6. Niche overlap within shelf communities.** (A) Mean posterior estimates of isotopic niche regions constructed from $\delta^{15}$N and $\delta^{13}$C values for benthic deposit and suspension feeders sampled in shelf coastscapes, in three geographic sectors. Niche regions were calculated at alpha = 0.95. (B) Density distribution of $\delta^{13}$C data from each sector, with dashed lines indicating the sample mean.

**Table 7. Quantification of isotopic niche overlap in shelves.**

| SHELF | Bering/Chukchi | Beaufort | Svalbard |
|---|---|---|---|
| **Bering/Chukchi** | NA | 77.6 (69.8, 84.8) | 42.6 (28.2, 60.3) |
| **Beaufort** | 67.8 (61.8, 73.7) | NA | 52.5 (36.9, 70.0) |
| **Svalbard** | 37.7 (28.8, 47.2) | 49.0 (36.6, 61.5) | NA |

Mean posterior estimates of overlap probability and 95% credible interval (CI) range for suspension and deposit feeders from each sector within shelves. Numbers represent the probability that an individual from the sector indicated in the row will be found within the niche indicated in the column. Values are reported as mean posterior estimate with lower and upper limits in parentheses (2.5% CI, 97.5% CI).

seasonality of primary production in an Arctic system [46]. However, we found that this overlap in isotopic niche space applies not only to feeding habits within a coastscape but also across geomorphologically very different coastscapes.

Evidence to refute the third hypothesis that isotopic niches for a common coastscape across longitudinal sectors would overlap was less clear. Although we generally found overlap in the isotopic niches between common coastscapes, we also found some regional distinctions. Fjords in the Canadian Arctic Archipelago, Eastern Greenland, and Svalbard exhibited overlapping isotopic niches on the $\delta^{13}$C axis, indicating mostly isotopically similar food sources were assimilated by consumers (Fig 5). However, the isotopic niche for consumers from the Canadian Arctic Archipelago extended to relatively $^{15}$N-enriched space, unlike those from Eastern Greenland and Svalbard. The greater depth in some of the Canadian Arctic Archipelago stations can facilitate depth-associated $^{15}$N-enrichment of sinking pPOM [61], expanding the isotopic niche compared to its counterparts.

While isotopic niche overlap occurred between suspension and deposit feeders in the Northern Bering/Chukchi Seas, Beaufort Sea, and Svalbard sectors, each isotopic niche extended into distinct isospace for both carbon and nitrogen (Fig 6). The distinctly $^{13}$C-depleted values in the Beaufort Sea (Fig 6; Table 7) align with the enormous export of terrestrial carbon

originating from the Mackenzie River in the eastern Beaufort and the Colville River in the western Beaufort [97] and inputs from the Alaska Coastal Water from the Chukchi Sea [32,98]. Evidence for consumer assimilation of these $^{13}$C-depleted terrestrial sources by consumers has been widely noted [34,48,99]; thus, our finding that the Beaufort Sea consumer isotopic niche extends into $^{13}$C-depleted isospace compared to other longitudinal sectors is not surprising. Conversely, the suspension and deposit feeders from the Svalbard shelves had δ$^{15}$N values that were more $^{15}$N-depleted compared to the Northern Bering/Chukchi Seas and Beaufort Sea sectors (Fig 6). These lower δ$^{15}$N values for shelf consumers were also reflected in pPOM (Table 3); mean δ$^{15}$N values for Northern Bering/Chukchi Seas (7.6±2.4‰) and Beaufort Sea (6.8±2.2‰) sector pPOM were~3‰ higher than in the Svalbard (4.4±2.5‰). We surmise this difference in pPOM δ$^{15}$N values is ultimately related to different sources of dissolved inorganic nitrogen assimilated by primary producers that comprise the particulate pools in these two regions. For example, upwelled nitrate from the Pacific Basin that flows northward to the Northern Bering and Chukchi Seas is $^{15}$N-enriched on average by 3‰ compared to relatively $^{15}$N-depleted nitrate that is delivered to the waters surrounding Svalbard from the Atlantic Basin [100]. This trend may be exacerbated in the coastal waters around Svalbard by ongoing "Atlantification", a term describing the increased influence of Atlantic Ocean heat and nutrients in the Atlantic Arctic region due to massive reduction in sea ice and a weakened halocline [101]. The δ$^{15}$N signature of the Beaufort Sea sector west of the Mackenzie River outflow is more intermediate, a reflection of both terrestrial inorganic-N sources from the Alaska Coastal Water and the advection of $^{15}$N enriched pPOM from Northern Bering Sea water that is advected eastward onto the Alaskan Beaufort Sea shelf as the Beaufort Undercurrent [98]. Northern Bering/Chukchi Seas sector pPOM was the most $^{15}$N-enriched of any sector, primarily derived from the aforementioned Bering Sea water.

## Organic matter resources assimilated in Arctic coastscapes

Following the tenets of stable isotope ecology, one should be able to infer the range of assimilated organic matter sources by the range of stable isotope values exhibited by consumers [24]. All feeding groups among pan-Arctic coastscapes contained consumers with δ$^{13}$C values between −24.0‰ and −15.4‰ (Table 3). The large δ$^{13}$C ranges for each of the four feeding groups among the four coastscapes (> 9‰) is a clear indicator of assimilation of multiple isotopically-distinct organic matter sources across the coastal Arctic.

Like consumers, pPOM and sPOM from the four coastscapes collectively reflected large δ$^{13}$C ranges: 12.7‰ (−33.5‰ to −20.8‰) in fjords; 9.8‰ (−31.5‰ to −21.7‰) in lagoons; 8.5‰ (−28.5‰ to −20.0‰) on shelves; and 11.5‰ (−31.3‰ to −19.8‰) in straits (Fig 2, Table 3). The wide range of δ$^{13}$C values for pPOM and sPOM indicated these end-member pools were indeed a conglomeration of multiple isotopically-distinct organic matter sources.

Phytoplankton are a common constituent of the pPOM mixture in most marine habitats and may sink and become incorporated into the sPOM pool [64,65,85]. Within the pPOM aggregation in this dataset, data rows specifically called phytoplankton ranged from −26.7‰ to −20.3‰ for δ$^{13}$C. This matches other reported values for coastal Arctic phytoplankton of −26‰ to −19‰ [92,93,102]. However, the range of phytoplankton stable isotope values (−26‰ to −19‰) cannot solely account for the range of pPOM (−33.5‰ to −19.8‰) and sPOM (−28.5‰ to −19.0‰) reported here. Therefore, other $^{13}$C-depleted carbon sources relative to phytoplankton must have contributed to the mixture of POM pools.

Potential sources typically $^{13}$C-depleted compared to marine phytoplankton include terrestrial plant material (−27.7±1.3‰) [18], thawed permafrost organic matter (−27.7±1.4‰) [18], freshwater microalgae (−33.1±4.7‰) [18], red macroalgae (−33.0±3.3‰) including common Arctic genera such as *Phycodrys*, *Polysiphonia*, and *Odonthalia* [15,18,60,103], and some brown macroalgae such as *Desmarestia* (<−27‰) [60]. Organic matter sources in Arctic coastscapes that can be $^{13}$C-enriched compared to phytoplankton are brown macroalgae like the kelp *Laminaria* and other macroalgae including *Fucus, Pylaiella, and Ectocarpus* (−22‰ to −14‰) [51,60,104], iPOM [80,92,96,105–107], and benthic microalgae [108–112].

The coastscapes that exhibit the required hard substrate for significant macroalgal production likely exist east of the Beaufort Sea in the straits and fjords of the Canadian Arctic Archipelago, Greenland, and Svalbard. Red and brown

macroalgae require hard benthic substrate (e.g., rocky shorelines, seafloor cobbles, etc.) and sufficient benthic PAR to subsist. Lantuit et al. [2] report that approximately one-third of the Arctic coastline is lithified. In addition, coastal areas impacted by rivers receive high levels of sediment inputs. Macroalgal carbon assimilation is less prevalent than other available sources where freshwater influence is pronounced [55,60].

iPOM carbon is a critical resource for Arctic consumers [63,113,114]. One recent pan-Arctic biomarker analysis demonstrated that not only was iPOM a carbon subsidy available year-round in sediment food banks, but most (133 of 155) species analyzed, including invertebrates and vertebrates, assimilated iPOM carbon to some degree [115]. Another study using the same biomarkers demonstrated fjord and shelf consumers near Young Sound, Greenland assimilated no less than 60% and oftentimes more than 90% iPOM-derived organic matter [83]. While this trend likely prevails throughout Arctic coastscapes, the stable isotopic evidence for its assimilation is not always clear (Fig 7). In some cases, for example on shelves, the $^{13}$C-enriched values of iPOM may explain some of the $^{13}$C-enriched values of consumers. Within the dataset assembled for this study, iPOM possessed $\delta^{13}$C values as high as −5.5‰ in some instances, but in other cases iPOM data were less $^{13}$C-enriched than phytoplankton (Fig 7). iPOM $\delta^{13}$C values ranged from −30.2‰ to −14.4‰ in fjords, from −28.4‰ to −25.0‰ in lagoons, and −37.0‰ to −5.5‰ on shelves (Table 3). The wide range in $\delta^{13}$C values is attributed to a variety of factors including PAR availability, temperature, $\delta^{13}$C-DIC, DIC limitation, nutrient concentration, and cell growth rates [116,117]. However, iPOM $\delta^{13}$C values observed in this dataset were not only variable but also overlapped with other organic matter sources in lagoons, fjords, and shelves, so natural abundance stable isotopes were too confounded to demonstrate the same widespread assimilation reported elsewhere [83,115].

Benthic microalgal carbon is recognized as a major source of labile carbon to consumers in non-Arctic coastal habitats, such as fjord-like areas in Antarctica [111], shallow temperate bays [109], and shallow continental shelves [108]. The shallow coastal habitats of the Arctic where PAR can penetrate to the seafloor [22] can foster populations of benthic microalgae, which would be a food source to the benthic food web [15,58,112,118]. These algae are known for their high relative contribution to total primary production rates in Arctic waters under 30 m depth compared to phytoplankton and ice algae [119] and can be observed through high sediment pigment concentrations [85,120–122], but direct measurement of stable isotope values for Arctic benthic microalgae remains elusive [58]. Modeled values of benthic microalgae $\delta^{13}$C values for the Beaufort Sea shelf are reported to lie between −24‰ and −18‰ [110], which falls within the range of a potential food source for shelf consumer compiled in this study (Figs 4 and 7).

$^{13}$C-enriched benthic primary producers like macroalgae, benthic microalgae, and ice algae contribute up to 30% of annual phytoplankton production at a pan-Arctic Ocean scale based on a light availability model [23], and may be proportionally more important in shallow coastal areas [120]. Deeper Arctic coastscapes like the straits of the Canadian Arctic Archipelago and deep fjords would not host benthic microalgae unless they were physically advected from shallow depths.

We acknowledge that the use of bulk stable isotopes, broad feeding habit categories, and wide geographic coverage leaves open the possibility that species-specific selective assimilation, isotopic routing between tissue types, and differences in isotopic discrimination could occur without detection by the methods employed here [123–125]. However, we argue the comparisons of bulk stable isotope values between consumers and organic matter sources described above are robust to individual or species-level instances of atypical stable isotope behavior due to the volume of aggregated data used to parse broad patterns.

## Missing pieces to the trophic puzzle

There still remain some gaps in our understanding of isotopic relationships and the relative importance of sampled end-members. While our evidence suggests that POM pools are becoming more $^{13}$C-depleted over decadal timescales (Fig 3), there are many consumers that exist in $^{13}$C-enriched isospace without an obvious connection to a $^{13}$C-enriched end-member (Fig 7). Typically, trophic enrichment factors of ~+1‰ are assumed for $\delta^{13}$C and +1–4‰ for $\delta^{15}$N [126]. Nadon and Himmelman [127] prescribed trophic enrichment factors of +4‰ for $\delta^{13}$C between primary producers and

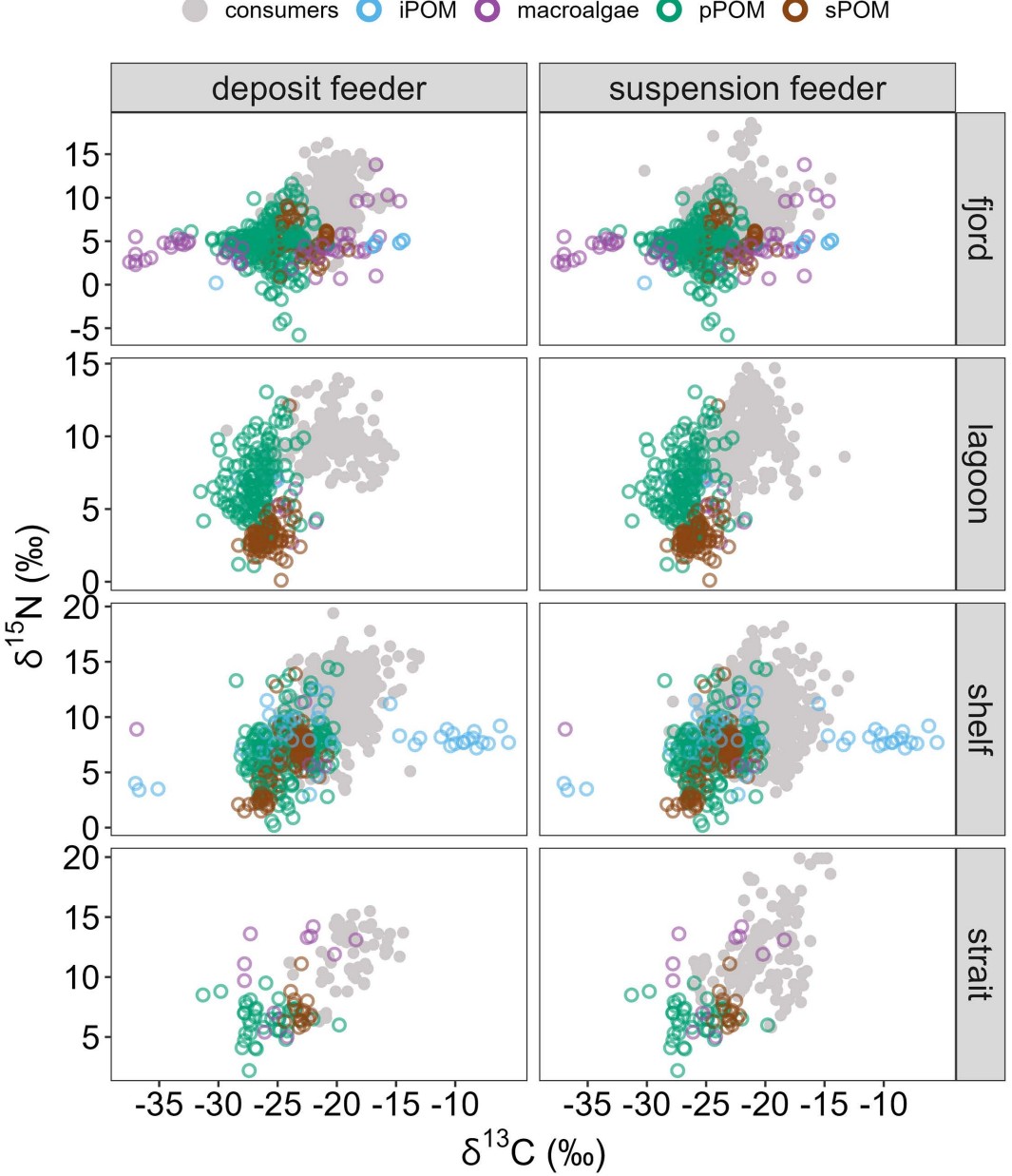

**Fig 7. Stable isotope biplots of end-members (colored points) faceted by coastscape and overlayed consumers (gray points).** Panels are faceted by feeding habit (deposit feeder or suspension feeder) and coastscape. Deposit and suspension feeders were plotted to depict primary consumers. Note the different y-axis scale for fjords. Predators and scavenger/omnivore data not shown.

primary consumers based on their observations, but even with this increased trophic step the end-members represented here, particularly in lagoons and straits, cannot account for the $^{13}$C-enriched values of consumers. Notably, the few iPOM values reported for lagoons were relatively $^{13}$C-depleted, and no iPOM data were reported for straits. We note the lack of data imported during this non-exhaustive literature survey is not evidence of absence for iPOM in these coastscapes, especially as sophisticated biomarker methods continually demonstrate the importance of iPOM to consumers [83,115]. Some iPOM stable isotope data from the Canadian Arctic Archipelago that failed to satisfy the metadata requirements for

inclusion in this study demonstrate iPOM can be a [13]C-enriched food source to the benthic food web [96,106,107]. The [13]C-depleted values of iPOM in lagoons is puzzling but not out of the range of other reported values [35,105,106]. [13]C-enrichment of sea ice algae often occurs within assemblages with high chlorophyll *a* concentrations and may occur later in the season when more light availability increases primary production rates [105,106].

Moreover, there could be a mismatch between end-member sampling and the isotopic turnover window reflected by the consumers if both are collected concurrently. The vast majority of samples represented in this pan-Arctic study were collected between mid-July and mid-September (S6 Fig). Since the isotopic turnover of Arctic invertebrates [128,129] and fish [130] can be quite slow, on the order of 1–2 months if not longer, some consumers collected during summer months may still isotopically reflect their diet from previous ice-covered months, like iPOM.

Lastly, it is unclear what role bacteria play in modifying organic matter sources prior to assimilation by the benthic food web or act as a food source themselves. Sinking organic matter has been documented to undergo stable isotope value changes from microbial breakdown before reaching the benthos [61,77]. This suggests organic matter would continue to undergo microbial breakdown once it arrives to the seafloor. Bacterial breakdown of organic matter has been invoked as a mechanism that results in an unmeasured [13]C-enriched food source for benthic consumers that exhibit $\delta^{13}C$ values heavier than sampled end-members in Arctic food webs [57,131,132]. Techniques more sophisticated than bulk stable isotope analysis are required to follow bacterially-derived carbon into the food web, and this approach indicates it is significantly important to Arctic consumers [133].

## Conclusion

This analysis of Arctic coastal stable isotope data from 1999–2022 showed long-term changes in food sources ($\delta^{13}C$ values of pPOM and sPOM) that may reveal the effects of warming trends across coastal Arctic ecosystems. The $\delta^{13}C$ values of pPOM and sPOM becoming more [13]C-depleted over the last two decades can be partially related to the Suess Effect where primary producers incorporate [13]C-depleted $CO_2$ derived from fossil fuel combustion. However, these isotopic temporal trends of the POM pools are most likely based on a combination of (1) increased incorporation of terrestrially-derived organic matter, (2) increased incorporation of autochthonous microalgae using [13]C-depleted terrestrially-derived DIC, or (3) decreased contributions from [13]C-enriched sympagic sources. This [13]C-depletion over two decades was also found in Arctic consumers. Across all Arctic coastscapes, consumers exhibited overlapping isotopic composition, notably with wide $\delta^{13}C$ ranges that indicated assimilation of multiple organic matter sources, including terrestrial organic matter, pPOM, sPOM, iPOM, macroalgae, and probably benthic microalgae except in the deep Canadian straits. This consistent pattern across coastscapes supports the notion that Arctic benthic invertebrate consumers have high trophic plasticity that allows them to use a wide variety of sources, which stabilizes food webs [56,134,135]. Lastly, consumers within the same coastscape exhibited overlapping isotopic niches between longitudinal sectors, with a few notable exceptions. Consumer isotopic niches reflected regional trends in freshwater and nutrient inputs that influence endmember isotope ranges, for example, the signature of increasing freshwater influence in the Beaufort Sea and the "Atlantification" near Svalbard represent regional effects of climate warming visible in the pan-Arctic isoscape.

## Supporting information

**S1 Fig. Linear regressions of consumer stable carbon and nitrogen isotope values and collection date.** Graphical display of regressions faceted by feeding habit and stable isotope for the linear regressions presented in Table 4. (PNG)

**S2 Fig. Posterior estimates of niche overlap for suspension feeders.** Probability that an individual sampled in the coastscape listed for the row would also occur within the isotopic niche of the same feeding guild samples in the coastscape listed for the column. (TIF)

**S3 Fig. Posterior estimates of niche overlap for deposit feeders.** Probability that an individual sampled in the coastscape listed for the row would also occur within the isotopic niche of the same feeding guild samples in the coastscape listed for the column.
(TIF)

**S4 Fig. Posterior estimates of niche overlap for opportunist/scavengers.** Probability that an individual sampled in the coastscape listed for the row would also occur within the isotopic niche of the same feeding guild samples in the coastscape listed for the column.
(TIF)

**S5 Fig. Posterior estimates of niche overlap for predators.** Probability that an individual sampled in the coastscape listed for the row would also occur within the isotopic niche of the same feeding guild samples in the coastscape listed for the column.
(TIF)

**S6 Fig. Histogram of day-of-year for sample collection for data collected for this meta-analysis.** Counts represent rows of data. End-members and consumers from all years included.
(PNG)

**S1 Table. Data sources used for meta-analysis.** Full bibliographic information available for each publication listed in References. Projects listed with acronyms represent datasets unpublished in peer-review literature. All data are available in public repository referenced in manuscript.
(DOCX)

**S2 Table. Genera in the meta-analysis dataset that were mathematically corrected for lipid content.** Corrections were from equations represented in Table 2.
(DOCX)

## Acknowledgments

We are grateful for the many captains, crews, and researchers that have worked over the past two decades to collect these samples from across the Arctic and produce data aggregated here. Without their countless hours of work, this synthesis would not be possible. We thank L. Ye and an anonymous reviewer for their feedback that improved this manuscript.

## Author contributions

**Conceptualization:** Nathan D. McTigue, Katrin Iken, Kenneth H. Dunton.

**Formal analysis:** Nathan D. McTigue, Ashley Ehrman.

**Funding acquisition:** Katrin Iken, Janne E. Søreide, Kenneth H. Dunton.

**Investigation:** Nathan D. McTigue.

**Methodology:** Nathan D. McTigue, Katrin Iken.

**Supervision:** Katrin Iken.

**Visualization:** Nathan D. McTigue, Ashley Ehrman.

**Writing – original draft:** Nathan D. McTigue.

**Writing – review & editing:** Nathan D. McTigue, Katrin Iken, Ashley Ehrman, Bodil A. Bluhm, Guillaume Bridier, Rolf Gradinger, Joanna Legeżyńska, Maeve McGovern, Bailey McMeans, Frédéric Olivier, Amanda Poste, Paul E. Renaud, Virginie Roy, Janne E. Søreide, Maria Włodarska-Kowalczuk, Kenneth H. Dunton.

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
