## [Decision Letter · Decision Letter 0]

28 May 2025

PONE-D-25-17749
Trophic niche variation across the Arctic coastal continuum
PLOS ONE

Dear Dr. McTigue,

Thank you for submitting your manuscript to PLOS ONE. After careful consideration, we feel that it has merit but does not fully meet PLOS ONE’s publication criteria as it currently stands. Therefore, we invite you to submit a revised version of the manuscript that addresses the points raised during the review process.

We look forward to receiving your revised manuscript.

Kind regards,

Vitor Hugo Rodrigues Paiva, Ph.D.

Academic Editor

PLOS ONE

Journal Requirements:

3. Thank you for stating the following financial disclosure: [This research was funded through the 2017-2018 Belmont Forum and BiodivERsA joint call for research proposals, under the BiodivScen ERA-Net COFUND program, and with the funding organizations National Science Foundation (#1906726), Research Council of Norway (#296836/31406), and National Science Center, Poland (NCN #31382). The work was also partially funded by the Beaufort Lagoon Ecosystems LTER (NSF #1656026 and #2322664) and the Arctic Marine Biodiversity Observing Network (NOAA #NA19NOS0120198, ONF #N00014-22-1-2793, NASA #80NSSC22K1780).]. 

Please state what role the funders took in the study.  If the funders had no role, please state: ''The funders had no role in study design, data collection and analysis, decision to publish, or preparation of the manuscript.''

5.Please amend either the title on the online submission form (via Edit Submission) or the title in the manuscript so that they are identical.

6. We note that Figure1 and Supplemental (striking image.tif)  in your submission contain [map/satellite] images which may be copyrighted. All PLOS content is published under the Creative Commons Attribution License (CC BY 4.0), which means that the manuscript, images, and Supporting Information files will be freely available online, and any third party is permitted to access, download, copy, distribute, and use these materials in any way, even commercially, with proper attribution. For these reasons, we cannot publish previously copyrighted maps or satellite images created using proprietary data, such as Google software (Google Maps, Street View, and Earth). For more information, see our copyright guidelines: http://journals.plos.org/plosone/s/licenses-and-copyright.

1. You may seek permission from the original copyright holder of Figure1 and Supplemental (striking image.tif)  to publish the content specifically under the CC BY 4.0 license. 

Please upload the completed Content Permission Form or other proof of granted permissions as an ''Other'' file with your submission.

Natural Earth (public domain): http://www.naturalearthdata.com

Reviewers' comments:

Reviewer's Responses to Questions

**Comments to the Author**

1. Is the manuscript technically sound, and do the data support the conclusions?

Reviewer #1: Yes

Reviewer #2: Partly

2. Has the statistical analysis been performed appropriately and rigorously? 

Reviewer #1: Yes

Reviewer #2: Yes

3. Have the authors made all data underlying the findings in their manuscript fully available?

Reviewer #1: Yes

Reviewer #2: Yes

4. Is the manuscript presented in an intelligible fashion and written in standard English?

Reviewer #1: Yes

Reviewer #2: Yes

5. Review Comments to the Author

Reviewer #1: This manuscript focuses on a highly relevant and ecologically important question: how trophic niche structures in pan-Arctic coastal ecosystems have changed under the dual influence of environmental change and coastal geomorphology. The topic fits well with current global research on Arctic carbon cycling and land–ocean interactions and has clear scientific value. The study compiles and analyses more than 10,000 stable isotope records across multiple trophic levels over the past two decades, showing its strength in data synthesis and regional pattern identification. A particularly commendable contribution is the development and use of the “coastscape” concept, which classifies Arctic coastal systems based on geomorphology, ecological function, and biogeochemical traits. This framework enables meaningful comparisons of consumer isotopic niches across fjords, lagoons, shelves, and straits. The study is a large undertaking with wide spatial and temporal coverage, and it uses appropriate methods such as non-parametric statistics and Bayesian niche overlap modelling (nicheROVER).

However, although the authors demonstrate solid academic ability in building the dataset and analytical framework, some key ecological processes are not discussed in sufficient depth. Generally, the paper does not fully explore the selective assimilation of organic matter by consumers, the role of remineralisation in explaining δ¹⁵N differences in sPOM, or the full pathway of isotopic signal transfer from source to sediment. Also, the assumed δ¹³C values for certain end-members, such as phytoplankton, are somewhat conservative and may lead to misjudging their contribution to consumers. The overall writing is generally clear, but some mechanism discussions lack depth in terms of biogeochemical processes. In summary, I recommend minor revision, as the conclusions are mostly supported by the data and the manuscript is suitable for publication in PLOS ONE after revisions.

Specific comments:

Line 48: Please shorten the abstract slightly. Focus on the main message and avoid separating the conclusions.

Lines 75–77: The phrase “situated in the crucible of climate change” is too rhetorical. Replace it with a more neutral and scientific expression. Similar long or redundant sentences appear elsewhere and should be trimmed for clarity.

Lines 143–146: This is a key sentence. Please highlight the novel aspects of this study, especially regarding trophic niche expression and ecological plasticity.

Line 210: The description of iPOM should be moved to here from Line 213 for better logical flow.

Lines 308–311: Please explain more clearly how “overlap probability” is calculated. A short mathematical description and interpretation would be helpful.

Line 359: Although regression results for δ¹³C and δ¹⁵N are reported as significant, model fit indicators (e.g. R² or confidence intervals for Kendall’s tau) should be added in Fig. 2 and the text to strengthen the interpretation.

Lines 490–493: Consider adding a rough estimate of freshwater aquatic microalgae input. Based on radiocarbon data, most POM from rivers may be degraded before reaching the coastal ocean (e.g. Goñi et al., 2013).

Lines 504–507: The explanation provided is questionable. In the Bering Strait and Chukchi Shelf, terrestrial OM is often more likely to remain suspended in surface waters than marine OM (e.g. Ye et al., 2024). The general statement that terrestrial OM preferentially enters sediments should be reconsidered in light of regional variability. Also, the δ¹⁵N difference between sPOM and pPOM should not be attributed solely to end-member differences. Please consider the influence of mineralisation in surface sediments, including nitrogen regeneration and microbial reworking, and explain why δ¹⁵N is more affected than δ¹³C.

Lines 513–522: The manuscript connects isotope values of end-members and consumers over time, but does not clearly describe a full mechanism for signal transfer from pPOM → consumer → sPOM. Please outline the steps of fractionation, assimilation, unassimilated residue, and sedimentation more clearly to strengthen the logic.

Line 598: Please recheck the δ¹³C range for Arctic coastal phytoplankton (−26‰ to −23‰), as this may be too low. Several studies report values of about −21‰ or higher in shallow productive waters (e.g. Søreide et al., 2006), and underestimating this range may lead to underestimating phytoplankton contributions.

Lines 599–601: Please discuss the possibility that consumers preferentially assimilate ¹³C-rich compounds (e.g. proteins), which could cause consumer δ¹³C values to increase while leaving behind ¹³C-depleted lipids in the remaining pPOM.

Reviewer #2: This study integrates two decades of pan-Arctic coastal ecosystem data to explore spatiotemporal changes in consumer isotope niches under climate change, with significant ecological and environmental implications. Three null hypotheses (stability of isotope values over decades, consistency of isotope niches between coastal landscapes, and regional differences within coastal landscapes) provide a clear analytical framework.

The significant decrease in δ13C values of pPOM/sPOM (2.1‰–2.2‰ per decade) and the parallel trends in consumers strongly support the hypothesis of increased terrestrial organic matter input, providing quantitative evidence for the trend collected in recent years at sampling sites in the high latitudes of the Arctic.Is the “terrestrialization”?

How have different landscapes at similar or close latitudes varied isotopically over the past two decades?

Comments

1. Add keywords;

2. Label place names in Figure 1;

3. Lines 556 to 561, 574 to 577

The influence of the Alaska Coastal Current on the distribution and transport of organic matter and its potential impact on isotopic signatures in coastal food webs should also be considered.

6. PLOS authors have the option to publish the peer review history of their article (what does this mean?). If published, this will include your full peer review and any attached files.

Reviewer #1: **Yes: **Liming Ye

Reviewer #2: No

---

## [Author Response · Author response to Decision Letter 1]

10 Jul 2025

The following response to the decision letter was uploaded in the "Attached Files" section with formatted text that would make it easier to read than in this text format. Regardless, the information is copied here as well.

Dear Vitor Hugo Rodrigues Paiva, Liming Ye, and the anonymous reviewer,

We thank you for your feedback and constructive comments that improved this manuscript. We feel the peer-review process has benefited us scientifically and improved the merits of this manuscript for publication. Below, we respond to each comment, both from the handling editor and the two reviewers. Our responses are in bold and blue text to facilitate reading the original comments (black typeface) and our responses. We also list the new line number(s) that changes were made when the comment required us to revise the text.

Best regards,

Nathan McTigue

Journal Requirements:

The style requirements have been checked. References have been checked for proper placement of special characters and sub/superscripting.

No author-generated code underpins the findings in this manuscript.

3. Thank you for stating the following financial disclosure: [This research was funded through the 2017-2018 Belmont Forum and BiodivERsA joint call for research proposals, under the BiodivScen ERA-Net COFUND program, and with the funding organizations National Science Foundation (#1906726), Research Council of Norway (#296836/31406), and National Science Center, Poland (NCN #31382). The work was also partially funded by the Beaufort Lagoon Ecosystems LTER (NSF #1656026 and #2322664) and the Arctic Marine Biodiversity Observing Network (NOAA #NA19NOS0120198, ONF #N00014-22-1-2793, NASA #80NSSC22K1780).]. BB contributed as part of the Kitikmeot Sea Science Study with funding provided by Fisheries and Oceans Canada, Polar Knowledge Canada, and the Arctic Research Foundation.

Please state what role the funders took in the study. If the funders had no role, please state: ''The funders had no role in study design, data collection and analysis, decision to publish, or preparation of the manuscript.''

The updated financial disclosure statement is:

This research was funded through the 2017-2018 Belmont Forum and BiodivERsA joint call for research proposals, under the BiodivScen ERA-Net COFUND program, and with the funding organizations National Science Foundation (#1906726), Research Council of Norway (#296836/31406), and National Science Center, Poland (NCN #31382). The work was also partially funded by the Beaufort Lagoon Ecosystems LTER (NSF #1656026 and #2322664) and the Arctic Marine Biodiversity Observing Network (NOAA #NA19NOS0120198, ONF #N00014-22-1-2793, NASA #80NSSC22K1780). BB contributed as part of the Kitikmeot Sea Science Study (K3S) with funding provided by Fisheries and Oceans Canada, Polar Knowledge Canada, and the Arctic Research Foundation. The funders had no role in study design, data collection and analysis, decision to publish, or preparation of the manuscript.

All data associated with this publication are published and freely available on the Environmental Data Initiative at https://doi.org/10.6073/pasta/48570e835463e405c707a4905c9d172d

5.Please amend either the title on the online submission form (via Edit Submission) or the title in the manuscript so that they are identical.

The title on the online submission form was amended.

6. We note that Figure1 and Supplemental (striking image.tif) in your submission contain [map/satellite] images which may be copyrighted. All PLOS content is published under the Creative Commons Attribution License (CC BY 4.0), which means that the manuscript, images, and Supporting Information files will be freely available online, and any third party is permitted to access, download, copy, distribute, and use these materials in any way, even commercially, with proper attribution. For these reasons, we cannot publish previously copyrighted maps or satellite images created using proprietary data, such as Google software (Google Maps, Street View, and Earth). For more information, see our copyright guidelines: http://journals.plos.org/plosone/s/licenses-and-copyright.

1. You may seek permission from the original copyright holder of Figure1 and Supplemental (striking image.tif) to publish the content specifically under the CC BY 4.0 license.

Please upload the completed Content Permission Form or other proof of granted permissions as an ''Other'' file with your submission.

The basemap in Fig. 1 was created using the R package ‘grfxtools’, which utilizes Natural Earth (public domain) to project map data. The map was further modified by the authors to show study sites. Thus, the map is within the public domain, is not copyrighted, and complies with the license of PLOS ONE. For more details, see https://github.com/EarthSystemDiagnostics/grfxtools/blob/main/LICENSE.note.

5. Review Comments to the Author

Reviewer #1:

This manuscript focuses on a highly relevant and ecologically important question: how trophic niche structures in pan-Arctic coastal ecosystems have changed under the dual influence of environmental change and coastal geomorphology. The topic fits well with current global research on Arctic carbon cycling and land–ocean interactions and has clear scientific value. The study compiles and analyses more than 10,000 stable isotope records across multiple trophic levels over the past two decades, showing its strength in data synthesis and regional pattern identification. A particularly commendable contribution is the development and use of the “coastscape” concept, which classifies Arctic coastal systems based on geomorphology, ecological function, and biogeochemical traits. This framework enables meaningful comparisons of consumer isotopic niches across fjords, lagoons, shelves, and straits. The study is a large undertaking with wide spatial and temporal coverage, and it uses appropriate methods such as non-parametric statistics and Bayesian niche overlap modelling (nicheROVER).

However, although the authors demonstrate solid academic ability in building the dataset and analytical framework, some key ecological processes are not discussed in sufficient depth. Generally, the paper does not fully explore the selective assimilation of organic matter by consumers, the role of remineralisation in explaining δ¹⁵N differences in sPOM, or the full pathway of isotopic signal transfer from source to sediment. Also, the assumed δ¹³C values for certain end-members, such as phytoplankton, are somewhat conservative and may lead to misjudging their contribution to consumers. The overall writing is generally clear, but some mechanism discussions lack depth in terms of biogeochemical processes. In summary, I recommend minor revision, as the conclusions are mostly supported by the data and the manuscript is suitable for publication in PLOS ONE after revisions.

We thank this reviewer for the attention to detail required to provide this constructive feedback that ultimately improved this manuscript. We address these concerns below.

Specific comments:

Line 48: Please shorten the abstract slightly. Focus on the main message and avoid separating the conclusions.

The abstract was shortened from 295 words to 255 by distilling the conclusions into overarching ideas.

Lines 75–77: The phrase “situated in the crucible of climate change” is too rhetorical. Replace it with a more neutral and scientific expression. Similar long or redundant sentences appear elsewhere and should be trimmed for clarity.

This phrase was removed and the sentence was edited to “This expanse of coastal ecosystems lining the Arctic Ocean are subjected to the impacts of warming-induced changes from both adjacent terrestrial and marine ecosystems.” Now L72.

Lines 143–146: This is a key sentence. Please highlight the novel aspects of this study, especially regarding trophic niche expression and ecological plasticity.

To highlight the novelty of the study we added the sentence “Synthesizing this information at the pan-Arctic level is a novel approach to investigate trophic niche variation and explore the ecological plasticity of consumers at the land-sea-ice interface of the coastal Arctic where multiple environmental changes are occurring in tandem.” Now L143-146.

Line 210: The description of iPOM should be moved to here from Line 213 for better logical flow.

Done.

Lines 308–311: Please explain more clearly how “overlap probability” is calculated. A short mathematical description and interpretation would be helpful.

The text now reads: “As mathematically described by Swanson et al. [30], the method uses a Bayesian framework to estimate the n-dimensional space (NR) that a data point has an α probability of being found, where n is the number of isotopes used and α is the user-defined probability, in this case 95%. Then probabilistic niche overlap between two groups is conducted by estimating the probability that a data point randomly selected from one group would fall within the NR space of the other group using a Monte Carlo technique. NR was informed from bivariate δ13C and δ15N values. The method is robust to differences in sample size, performs well with large sample sizes, and, unlike geometric methods, accounts for species-specific distributions in bivariate space when estimating the probability of overlap [30]. Uncertainty was accounted for by using a Bayesian approach with a noninformative normal-inverse-Wishart prior distribution and 10,000 random permutations to calculate the posterior mean and 95% credible intervals of probabilities of niche overlap.” L305-316.

Line 359: Although regression results for δ¹³C and δ¹⁵N are reported as significant, model fit indicators (e.g. R² or confidence intervals for Kendall’s tau) should be added in Fig. 2 and the text to strengthen the interpretation.

On L352-357 the text has been updated to “The linear regressions between collection date and pPOM δ13C values (p<0.001, Kendall's τ = -0.267) and collection date and sPOM δ13C values (p<0.001, Kendall’s τ = -0.446) revealed significant negative trends with slopes indicating decadal decreases of -2.1 and -2.2 ‰, respectively (Fig 3; Table 4). Likewise, the trend between collection date and sPOM δ15N values (p<0.001, Kendall’s τ = -0.478) was significantly negative with a decadal change of -3.1 ‰ per decade (Table 4).”

The caption for Fig 3 was also updated accordingly.

Lines 490–493: Consider adding a rough estimate of freshwater aquatic microalgae input. Based on radiocarbon data, most POM from rivers may be degraded before reaching the coastal ocean (e.g. Goñi et al., 2013).

Good point. Using the same radiocarbon approach described by Goñi et al. (2013), Behnke et al. (2023) estimated 39-60% of exported river organic matter is derived from aquatic algae. This clarification was added to the text on L499.

The text now reads: “Freshwater aquatic microalgae, with δ13C values of -33.1±4.7‰, constitute 39-60% of the POM exported by rivers to coastal environments [18].”

Lines 504–507: The explanation provided is questionable. In the Bering Strait and Chukchi Shelf, terrestrial OM is often more likely to remain suspended in surface waters than marine OM (e.g. Ye et al., 2024). The general statement that terrestrial OM preferentially enters sediments should be reconsidered in light of regional variability. Also, the δ¹⁵N difference between sPOM and pPOM should not be attributed solely to end-member differences. Please consider the influence of mineralisation in surface sediments, including nitrogen regeneration and microbial reworking, and explain why δ¹⁵N is more affected than δ¹³C.

Good point. Given the results of Ye et al. (2024), we can add this caveat to the mechanisms of OM entering sediments. This paragraph was substantially revised to include caveats and further speculation on the mechanism causing the d15N difference between sPOM and pPOM as mentioned in this comment and the following one. We considered regeneration and remineralization, but we suspect this would cause 15N-enrichment in the sPOM and not 15N-depletion (Montoya 2008). Regardless, further elaboration is given to this surprising finding.

Montoya, J. (2008). Nitrogen Stable Is

---

## [Decision Letter · Decision Letter 1]

12 Oct 2025

Trophic niche variation across the pan-Arctic coastal continuum

PONE-D-25-17749R1

Dear Dr. McTigue,

We’re pleased to inform you that your manuscript has been judged scientifically suitable for publication and will be formally accepted for publication once it meets all outstanding technical requirements.

Kind regards,

Vitor Hugo Rodrigues Paiva, Ph.D.

Academic Editor

PLOS ONE

Additional Editor Comments (optional):

Reviewers' comments:

Reviewer's Responses to Questions

**Comments to the Author**

1. If the authors have adequately addressed your comments raised in a previous round of review and you feel that this manuscript is now acceptable for publication, you may indicate that here to bypass the “Comments to the Author” section, enter your conflict of interest statement in the “Confidential to Editor” section, and submit your "Accept" recommendation.

Reviewer #2: All comments have been addressed

2. Is the manuscript technically sound, and do the data support the conclusions?

Reviewer #2: Yes

3. Has the statistical analysis been performed appropriately and rigorously? 

Reviewer #2: Yes

4. Have the authors made all data underlying the findings in their manuscript fully available?

Reviewer #2: Yes

5. Is the manuscript presented in an intelligible fashion and written in standard English?

Reviewer #2: Yes

6. Review Comments to the Author

Reviewer #2: (No Response)

7. PLOS authors have the option to publish the peer review history of their article (what does this mean?). If published, this will include your full peer review and any attached files.

Reviewer #2: No

---

## [Editor Report · Acceptance letter]

PONE-D-25-17749R1

PLOS ONE

Dear Dr. McTigue,

I'm pleased to inform you that your manuscript has been deemed suitable for publication in PLOS ONE. Congratulations! Your manuscript is now being handed over to our production team.

Kind regards,

on behalf of

Dr. Vitor Hugo Rodrigues Paiva

Academic Editor

PLOS ONE